# Comprehensive Serum Glycopeptide Spectra Analysis Combined with Machine Learning for Early Detection of Lung Cancer: A Case–Control Study

**DOI:** 10.3390/cancers17091474

**Published:** 2025-04-27

**Authors:** Koji Yamazaki, Shigeto Kawauchi, Masaki Okamoto, Kazuhiro Tanabe, Chihiro Hayashi, Mikio Mikami, Tetsuya Kusumoto

**Affiliations:** 1Department of Thoracic Surgery, National Hospital Organization Kyushu Medical Center, Chuo-ku, Fukuoka 810-0065, Japan; 2Department of Pathology, Clinical Research Centre, National Hospital Organization Kyushu Medical Centre, Chuo-ku, Fukuoka 810-0065, Japan; 3Department of Respirology, National Hospital Organization Kyushu Medical Center, Chuo-ku, Fukuoka 810-0065, Japan; 4Medical Solution Promotion Department, Medical Solution Segment, LSI Medience Corporation, Itabashi-ku, Tokyo 174-8555, Japan; 5Department of Medical Sciences, Shonan University of Medical Sciences, Yokohama 244-0806, Japan; 6Chigasaki Central Hospital, Women’s Center, Chigsaki 253-0041, Japan; 7Department of Gastrointestinal Surgery and Clinical Research Institute Cancer Research Division, National Hospital Organization Kyushu Medical Center, Chuo-ku, Fukuoka 810-0065, Japan

**Keywords:** lung cancer, biomarkers, machine learning, neural network, Comprehensive Serum Glycopeptide Spectra Analysis, α1-antitrypsin, α2-macroglobulin, glycosylation

## Abstract

Lung cancer is a leading cause of death worldwide. Traditional diagnostic methods like computed tomography are costly and involve radiation exposure, making them unsuitable for screening. Blood-based diagnostics offer a safer and more affordable alternative, helping enable earlier detection and treatment to improve patient survival. This study enrolled 199 patients with lung cancer and 590 healthy volunteers, and we analyzed nine tumor markers and enriched glycopeptides (EGPs) obtained from serum proteins using liquid chromatography–mass spectrometry in the individuals. We found that α1-antitrypsin and α2-macroglobulin with fully sialylated biantennary glycan could significantly distinguish between patients with lung cancer and healthy individuals. Comprehensive Serum Glycopeptide Spectra Analysis, integrating nine tumor markers and 1688 EGPs using a machine learning model, enhanced diagnostic accuracy and achieved an ROC-AUC score of 0.935. This method represents a significant advancement in cancer diagnostics, combining multiple biomarkers with cutting-edge machine learning to improve the early detection of lung cancer.

## 1. Introduction

Lung cancer is one of the most common types of cancer. It affects millions of individuals worldwide every year, and is a leading cause of cancer-related deaths globally. In 2020, an estimated 2.2 million new lung cancer cases and 1.8 million lung cancer-related deaths were reported worldwide [1]. Approximately 126,000 new cases of lung cancer and 75,000 deaths are annually reported in Japan alone [2]. The early detection of lung cancer can significantly improve survival rates (60–80%), as the five-year survival rate plummets to just 6–8% when cancer is diagnosed at a late stage [2,3]. Despite current medical advancements, lung cancer often leads to death when detected at advanced stages, largely because of its subtle symptoms and the limitations of traditional diagnostic methods. This highlights the critical need for more effective screening methods to identify lung cancer at earlier and more treatable stages.

Computed tomography (CT) is a highly effective tool for early lung cancer detection [4]; however, its widespread use is limited by its significant physical burden on patients and its high cost. Additionally, the procedure requires specialized training, making it less accessible, especially in areas where trained professionals are scarce. Therefore, CT is not a feasible option for the routine screening of large populations. Blood tests, on the other hand, present a non-invasive and cost-effective alternative that can greatly improve early detection and survival outcomes. Tumor markers in the blood, such as carcinoembryonic antigen (CEA) [5] and cytokeratin 19 fragment antigen 21-2 (CYFRA) [6], play a significant role in lung cancer diagnosis and monitoring. They are commonly used for assessing treatment effectiveness and monitoring the potential recurrence of lung cancer. However, their utility in early detection is limited, and they are generally used alongside other diagnostic methods. Researchers have investigated various bloodstream biomarkers for the early detection of lung cancer [7,8], including tumor-secreted proteins [9], microRNAs [10], DNA methylation in cell-free DNA [11], and exosomes [12]. However, the use of single markers for cancer diagnosis has shown limited efficacy, leading to the emergence of approaches that combine multiple markers [13]. Significant progress has been achieved by applying machine learning techniques [14] such as deep learning and ensemble learning [15,16] to handle these multiple markers. Additionally, the development of comprehensive analytical methods such as proteomics [17,18] and metabolomics [19,20] has further accelerated this trend. These advanced methodologies enable the integration of large biomarker datasets, enhancing the predictive power of diagnostic models and contributing to the development of more effective strategies for early cancer detection.

It has been well-established that the aberrant glycosylation of serum proteins occurs alongside cancer development [21,22,23]. In particular, the aberrant glycosylation of haptoglobin induced by lung cancer development has been well studied [24]. However, the precise analysis of these glycan modifications is still challenging owing to multiple technical and methodological obstacles: (i) difficulty in obtaining specific antibodies against aberrant glycans; (ii) lack of intrinsic fluorescence or ultraviolet absorption properties in glycans; (iii) limited ionization efficiency, which hampers mass spectrometry-based analysis; (iv) necessity to enzymatically release glycans from host proteins before comprehensive analysis; and (v) laborious preparation and high cost associated with lectin assays, hindering the ability to administer large numbers of sample treatments. In our previous studies, we addressed these challenges by adopting a proteomic approach that focused on the analysis of glycopeptides rather than only glycans. This method could not only identify glycan alterations but also detect changes in serum protein expression associated with cancer progression [25]. Nonetheless, conventional glycoproteomic approaches face difficulties in effectively distinguishing glycopeptides from non-glycosylated peptides. Although lectins are commonly employed for this purpose, their specificity is limited—no single lectin can comprehensively recognize all glycan structures. Moreover, handling large numbers of samples using lectins requires significant amounts of labor. In this study, we focused on ensuring practical applicability and sought to establish a method that is simple, reliable, and suitable for high-throughput analysis by addressing two major challenges. First, glycopeptides were enriched based on their differences in molecular weight. When digested with trypsin, the molecular size of glycopeptides was significantly larger than that of non-glycosylated peptides and could be concentrated using ultrafiltration membranes [25]. While this approach led to partial contamination with large, non-glycosylated peptides, it significantly enhanced analytical throughput relative to lectin-based enrichment methods. Additionally, instead of identifying all glycopeptides detected by mass spectrometry, we focused exclusively on those that demonstrated significance within the discriminative model [25]. We employed this targeted strategy because identifying glycopeptides is inherently more difficult than identifying peptides in proteomic analyses, primarily due to the structural complexity of glycans [26]. These decisions substantially enhanced the efficiency of our analysis, enabling us to test more than 1000 cases [27].

The present study had two primary objects: (i) to identify novel cancer markers from over 10,000 enriched glycopeptides (EGPs) and (ii) to develop a machine learning model capable of detecting lung cancer. The model, known as Comprehensive Serum Glycopeptide Spectra Analysis (CSGSA), analyzes more than 1000 EGPs, integrates conventional tumor markers, and analyzes them using machine learning [27,28,29,30]. CSGSA is anticipated to significantly reduce both false positives and false negatives by combining clinically validated tumor markers, which have limited sensitivity, with glycans that are highly responsive to cancer onset. To assess the practical efficacy of our newly developed model for screening purposes, we evaluated its performance using the receiver operating characteristic area under the curve (ROC-AUC) and a positive predictive value (PPV) adjusted based on patient morbidity rates.

## 2. Materials and Methods

### 2.1. Study Design

This study was designed as a retrospective observational case–control analysis. All patients who presented to the hospital during the study period were considered for inclusion. Random sampling and blinding procedures were not applied. Based on an assumed alpha level of 0.05 and beta level of 0.2, and anticipating that the mean expression levels of the target markers would differ by roughly half a standard deviation between the cancer and healthy groups, the minimum required sample size was calculated to be approximately 100.

We enrolled 199 patients with lung cancer, and 590 healthy volunteers (Table 1). Serum samples were collected from the patients at the time of cancer detection and before any treatment or surgery. Serum samples of Japanese patients were obtained from our hospital, National Hospital Organization Kyushu Medical Center (Fukuoka, Japan), while those of Caucasian patients with cancer were obtained from KAC Corporation (Kyoto, Japan) and Sanfco Ltd. (Tokyo, Japan). Sera of healthy Japanese volunteers were obtained from LSI Medience Corporation (Tokyo, Japan) and SOIKEN (Osaka, Japan), while those of Caucasian, African American, and Hispanic volunteers were obtained from KAC Corporation and Sanfco Ltd. Informed consent was obtained from all patients and volunteers, and the use of patient clinical information and serum samples was approved by the Institutional Review Boards of National Hospital Organization Kyushu Medical Center (IRB registration number: 17C299; 20 December 2017) and LSI Medience Corporation (IRB registration number: MS/Shimura 17–19; 22 January 2018). All research methods and procedures were conducted in strictly accordance with the ethical principles outlined in the Declaration of Helsinki and in full compliance with the relevant institutional, national, and international guidelines.

The specific inclusion criteria for this study were as follows: (i) patients diagnosed with primary cancer through imaging or histological analysis; (ii) patients at initial either diagnosis or experiencing a recurrence of cancer without starting treatment yet; and (iii) patients aged 20 years or older at the time of consent. The exclusion criteria were as follows: (i) patients with severe renal, hepatic, respiratory, or cardiac dysfunction, or concurrent infectious diseases; (ii) patients deemed unsuitable for study enrollment by the attending physician; and (iii) patients who had already begun any form of treatment. Cancer staging was performed according to the TNM classification system provided by the Union for International Cancer Control (UICC) [31]. Blood samples were collected via venous puncture prior to surgery or treatment. The serum was separated from blood cells by centrifugation within 8 h and stored at −80 °C until analysis. All samples were only analyzed once because we had previously validated their accuracy and reproducibility [27].

### 2.2. Tumor Marker Analyses

Nine tumor markers, alpha-fetoprotein (AFP), carcinoembryonic antigen (CEA), carbohydrate antigen 19-9 (CA19-9), cytokeratin 19 fragment (CYFRA), cancer antigen 125 (CA125), prostate specific antigen (PSA), cancer antigen 15-3 (CA15-3), NCC-ST-439, and squamous cell carcinoma antigen (SCC antigen), were all analyzed by LSI Medience Corporation, a clinical testing laboratory (Tokyo, Japan). The selection of these nine tumor markers was based on two considerations: the need to accurately distinguish lung cancer from other cancer types and to enable future application of this approach to cancers beyond lung cancer.

### 2.3. Sample Preparation and Liquid Chromatography–Tandem Mass Spectrometry

The sample preparation and analysis methods were adapted from those outlined in our previous work [30], with the following modifications: 20 μL of serum was mixed with 120 μL of acetone, containing 10% trichloroacetic acid, to precipitate proteins. The protein precipitate was resuspended in a denaturing mixture consisting of 80 μg of urea (Wako Pure Chemical Industries), 100 μL of Tris-HCl buffer (pH 8.5), 10 μL of 0.1 M EDTA, 5 μL of 1 M Tris(2-carboxyethyl) phosphine hydrochloride (Sigma-Aldrich, St. Louis, MO, USA), and 38 μL of water. The proteins were denatured by incubating the solution at 37 °C for 10 min. Subsequently, 40 μL of 1 M 2-iodoacetamide solution (Wako Pure Chemical Industries) was added to alkylate thiol groups in the proteins. They were kept for 10 min at 37 °C in dark conditions. The mixture was then transferred to a 30 kDa ultrafiltration tube (Amicon Ultra 0.5 mL, Millipore Corp., Burlington, MA, USA) to remove the denaturing agents. Protein digestion was performed on the filter using 200 μL of 0.1 M Tris–HCl buffer (pH 8.5), 20 μL of 0.1 μg/μL trypsin (Wako Pure Chemical Industries, Osaka, Japan), and 20 μL of 0.1 μg/μL lysyl endopeptidase (Fujifilm Wako Pure Chemical Industries). The mixture was then incubated for 16 h at 37 °C. After digestion, the mixture was centrifuged at 11,500× *g* for 30 min. The resulting filtrate, which contained both digested peptides and glycopeptides, was then transferred to a 10 kDa ultrafiltration tube (Amicon Ultra 0.5 mL, Millipore Corp.) to separate glycopeptides from non-glycosylated peptides [25]. The compounds retained by 10 kDa ultrafiltration—referred to as enriched glycopeptides (EGPs)—were subsequently subjected to analysis using liquid chromatography coupled with quadrupole time-of-flight mass spectrometry (LC-QTOF-MS; HP1200 + 6540, Agilent Technologies, Palo Alto, CA, USA). The apparatus was equipped with a C18 column (Inertsil ODS-4, 2 μm, 100 Å, 100 mm × 2.1 mm ID, GL Science, Tokyo, Japan). The EGPs were eluted using a gradient program at a flow rate of 0.2 mL/min and a temperature of 40 °C: we started with 15% to 30% mobile phase B for the first 7 min. This was increased to 30% to 50% mobile phase B from 7 to 12 min, followed by a 2 min hold at 100% mobile phase B. Mobile phase A consisted of 0.1% formic acid in water, whereas mobile phase B consisted of 0.1% formic acid in 9.9% water and 90% acetonitrile. The mass spectrometer was operated in negative ion mode with a capillary voltage of 4000 V. All samples were analyzed once. In total, 1688 EGPs were chosen from more than 30,000 detected peaks using the following three steps: (i) removing low-reproducibility peaks (CV > 50%), (ii) removing low-reliability peaks (S/N < 5), and (iii) removing isotopes, adducts, and fragment ions (Appendix A). Subsequently, residual 1688 EGP peaks were used for biomarker screening and CSGSA diagnostics.

### 2.4. Data Processing

The methods used for data processing have been described in detail in our previous studies [25,26]. Briefly, the liquid chromatography–mass spectrometry (LC-MS) raw data were exported in a CSV format using the Mass Hunter Export software (B.07.00; Agilent Technologies). Using R (R 3.2.2; R Foundation for Statistical Computing, Vienna, Austria), we extracted the peak positions (retention times and *m*/*z* values) and peak areas. The Marker Analysis software (ver.03-04-2018), provided by LSI Medience Corporation (Tokyo, Japan), was then employed to align all peak areas, minimize noise, and correct any discrepancies [25]. The tolerances for *m*/*z* and retention time during peak alignment and assignment were maintained at 0.06 Da and 0.3 min, respectively. For each sample, the relative expression of 1688 EGPs was determined by calculating the expression ratios relative to a quality control standard. The samples were randomly split into two sets: 70% for training and 30% for testing. The model was then trained on the training set, and its accuracy was subsequently evaluated on the test set. This process was iterated 10 times to ensure the robustness of the model. The results from each iteration were aggregated, and the receiver operating characteristic-area under curve (ROC-AUC) was calculated from the cumulative results. The predicted output values of the model ranged from 0 to 1; these were then transformed into Comprehensive Serum Glycopeptide Spectra Analysis (CSGSA) scores using the following formula:CSGSA score = −log_10_ (1 − predicted value)

This conversion facilitates a more interpretable score for clinical and diagnostic use.

### 2.5. Identification of the Glycopeptides Contributing to Lung Cancer Discrimination

To identify glycopeptide structures, we compared the retention times, single mass spectra, and tandem mass spectrometry (MS/MS) patterns of the target glycopeptides with those of commercially available purified human serum proteins. These proteins included alpha-1-acid glycoprotein, complement C8, complement C9, complement factor H, fibrinogen, haptoglobin, alpha-2-macroglobulin, antitrypsin, and transferrin, all of which were purchased from Sigma-Aldrich (St. Louis, MO, USA). Following digestion, the glycopeptides from patients with lung cancer were analyzed. Matching the retention times, mass spectra, and MS/MS patterns between the patient-derived glycopeptides and the standards allowed us to identify the source of the glycopeptides. Considering all possible glycan structures and peptide sequences to which glycans could bind, all possible glycopeptide molecular weights were calculated and compared with the detected glycopeptide molecular ion peaks. The glycopeptide structure was confirmed when its theoretical molecular weight matched the observed value within a 0.03 Da tolerance.

### 2.6. Statistical Analysis

We developed a machine learning model using Python (version 3.12, 64-bit). To compare the levels of EGPs between patients with LGC and healthy controls, we employed Student’s *t*-test, assuming a parametric distribution for all EGPs. Missing data, primarily values below the detection threshold, were replaced with zero. SPSS (version 27.0, Chicago, IL, USA) and other proprietary software were used for comprehensive statistical analyses [25]. Principal component analysis (PCA) was conducted using the SIMCA software (version 13.0.3; Umetrics, Umeå, Sweden).

## 3. Results

### 3.1. Comparison of the Levels of Tumor Markers in Patients with LGC and Healthy Volunteers

Figure 1A presents the levels of nine tumor markers in patients with lung cancer (LGC) and healthy volunteers. All values were transformed using a logarithmic scale, and ROC analysis was conducted between the LGC and healthy groups. ROC curves with AUCs exceeding 0.7, indicating high diagnostic potential, are shown in blue. CEA and CYFRA levels were markedly elevated in patients with LGC, with AUCs of 0.798 and 0.806, respectively. In contrast, squamous cell carcinoma (SCC)-antigen (AUC = 0.684), carbohydrate antigen 19-9 (CA19-9, AUC = 0.620), and cancer antigen 125 (CA125, AUC = 0.636) showed slight responses in patients with LGC. The AUC of SCC antigen, a tumor marker primarily associated with squamous cell carcinoma [32], reached 0.800 specifically for squamous cell lung cancer, while it dropped to 0.664 for lung adenocarcinoma. While these markers alone are insufficient for reliable LGC identification, their combined use with CEA, CYFRA, and EGPs could potentially enhance both the sensitivity and specificity of LGC detection.

### 3.2. Volcano Plot Analysis, Principal Component Analysis, and Heatmap Analysis of EGPs in LGC

We extracted 1688 EGPs from the sera of patients with LGC as robust and reliable markers. Volcano plots revealed that EGPs in patients with LGC, compared with healthy individuals, underwent dramatic changes (Figure 1B). In contrast, principal component analysis (PCA) demonstrated that the distributions of the healthy and LGC groups overlapped, suggesting that changes in serum glycans associated with lung cancer development are limited (Figure 1C). Heatmap analysis also did not show any notable differences between the healthy and LGC groups (Figure 1D).

### 3.3. Identification of Novel LGC-Specific Biomarkers in EGPs

In our comprehensive analysis of nearly 10,000 EGPs, we identified promising candidate biomarkers that showed significant differences between patients with LGC and healthy controls. Specifically, candidates were selected based on extremely low p-values (below 10^−10^) derived from Student’s t-tests and a mean-fold change exceeding 1.5. To ensure accurate quantification, EGP expression levels were normalized against transferrin levels, which serve as a reliable endogenous internal standard because of their stable expression across samples. This minimizes the variability inherent in sample collection and preparation. After rigorous testing for operational reproducibility, markers with inconsistent performances were eliminated. We identified two glycopeptides, α1-antitrypsin (AT) and α2-macroglobulin (MG), that robustly differentiate patients with LGC from healthy individuals. The analysis of their glycan modifications and attachment sites revealed specific impacts on the ROC-AUC; notably, glycan chains attached to asparagine (Asn) at position 271 on antitrypsin and position 70 on macroglobulin significantly influenced diagnostic discrimination (Figure 2A). Although several glycans were detected on both AT and MG, only fully sialylated biantennary glycans and those with core fucose were evaluated, as these provided reliable quantitative measures. Although the differences between the two glycopeptides were minimal, the ROC-AUC of the fully sialylated biantennary glycan slightly exceeded that with core fucose. AT with fully sialylated biantennary glycan attached to asparagine 271 (AT271-FSG) reached an AUC of 0.758, and MG with fully sialylated biantennary glycan attached to asparagine 70 (MG70-FSG) reached an AUC of 0.742 when comparing the LGC group with the healthy group, demonstrating lower performances than CEA and CYFRA (Figure 2B). Figure 2C illustrates the relationship between either AT271-FSG or MG70-FSG and tumor markers, including CEA and CYFRA, which exhibit significant responses to LGC. Although slight positive correlations were observed, they were not strong, suggesting that the combination of these markers could further enhance the diagnostic accuracy of LGC.

### 3.4. Combination Analysis of Tumor Markers, AT271-FSG, MG70-FSG, and 1688 EPGs

To enhance diagnostic accuracy, we developed a comprehensive machine learning model that integrates conventional tumor markers with AT271-FSG and MG70-FSG, along with 1688 EGPs (Figure 2D). We assessed three models to elucidate the distinct contributions of these biomarkers: Model 1 used only nine tumor markers, Model 2 added AT271-FSG and MG70-FSG to these nine markers, and Model 3 included all markers used in Model 2 along with 100 key features obtained from 1688 EGPs processed using PCA. AT271-FSG and MG70-FSG, the two glycopeptides with the most effective responses, were modeled separately from the 1688 glycopeptides to clarify their individual contributions. The features were reduced from 1688 to 100 using PCA to mitigate overfitting, a frequent challenge in machine learning [33]. We developed a model using 70% of the randomly selected samples as the training set and evaluated its performance by using the remaining 30% as the test set. This process was repeated 10 times to ensure robustness, and the aggregated results were then analyzed using ROC analysis (Figure 3). While developing Model 3, we evaluated both XGBoost and neural network architectures. XGBoost, an advanced implementation of gradient boosting algorithms, excels in classification and regression tasks by creating an ensemble of weak prediction models, typically decision trees [34]. On the other hand, neural networks employ layers of interconnected nodes to model complex patterns in data and classify samples [35].

Figure 4A illustrates the architecture of the neural network, which includes an input layer, two pairs of dense and dropout layers, and an output layer. Overfitting was mitigated by limiting the number of dense layers to two and incorporating dropout layers. Figure 4B illustrates the typical tree structure optimized by XGBoost, where CEA, MG70-FSG, CYFRA, and AT271-FSG played crucial roles in determining LGC. The tree structure varied depending on the choice of the training set, affecting the structure of the lower layers in particular. A comparison of the performance of the two models showed that the neural network model outperformed XGBoost in detecting LGC (*p* = 0.027, Figure 4C). Consequently, the ROC-AUC scores for the models differentiating cancer groups from healthy groups were 0.819 (Model 1), 0.843 (Model 2), and 0.935 (Model 3), significantly outperforming the current tumor markers (Figure 4D,E). We further transformed the values predicted using Model 3 into CSGSA scores (ranging from 0 to 10) using the following equation:CSGSA score = −log_10_(1 − Model 3 predicted value).

Figure 4F presents histograms based on the CSGSA scores for LGC. CSGSA effectively differentiated patients with LGC from healthy individuals, with an ROC-AUC value of 0.935. A cutoff value of 1 was employed to classify samples into positive and negative groups, achieving a sensitivity of 57.2% for LGC, while the specificity exceeded 98.0% for the healthy group (Table 2). This cutoff value was strategically set to minimize false positives rather than false negatives, aiming to enhance the PPV, which is crucial for an effective screening test. The PPV for LGC was 2.8%. The lower PPVs, despite high specificity, were attributed to the low prevalence of LGC.

### 3.5. Relationship Between LGC Histological Type and CSGSA Score

Lung cancers are primarily classified into adenocarcinoma (50–60%), squamous cell carcinoma (20–30%), large-cell carcinoma (5–10%), and small-cell carcinoma (15–20%) [36]. In this study, we were able to collect a sufficient number of adenocarcinoma and squamous cell carcinoma cases and then compared the CSGSA scores of these two types of cancer. Figure 5A shows a histogram of the CSGSA scores for adenocarcinoma and squamous cell carcinoma, along with the corresponding ROC curves compared with healthy subjects. The results demonstrated a stronger response in squamous cell carcinoma, with a higher ROC-AUC value of 0.946, compared with adenocarcinoma (0.873). Visualizing the glycan expression patterns, we used uniform manifold approximation and projection (UMAP) to compare the distribution of the subjects. UMAP is a dimensionality reduction technique that simplifies complex, high-dimensional data into a lower-dimensional space, making it easier to visualize patterns and relationships within the data [37]. The results indicated that the distribution pattern of patients with adenocarcinoma closely resembled that of healthy subjects, whereas that of squamous cell carcinoma showed a slight difference (Figure 5B).

### 3.6. CSGSA Determination for Patients with Benign Lung Disease

CSGSA scores of six patients with non-cancerous lung diseases, including benign tumor, inflammatory lung disease, and pulmonary fibrosis, were obtained. The results showed that all scores were approximately 2 and did not exhibit the marked increase observed in patients with LGC. However, these scores were slightly higher than those of healthy subjects, whose values were typically around 1 or lower. Therefore, a much larger sample size is required than just six cases to accurately assess the occurrence of false positives in patients with benign lung diseases (Figure 5C).

### 3.7. Relationship Between CSGSA Score and Cancer Development (Stage)

We analyzed the relationship between CSGSA scores and cancer stage. When the CSGSA scores were categorized into five stages (0, I, II, III, and IV), the scores progressively increased with advancing stage (Figure 5D). Compared with the healthy group, patients with stage I had an ROC-AUC value of 0.914 for LGC, indicating the potential of the CSGSA test for detecting early-stage cancers.

### 3.8. Relationship Between CSGSA Scores and Ethnicity

Figure 6A presents the relationship between ethnicity and CSGSA scores. No significant differences were observed between Asian and Caucasian people; in healthy individuals, scores were mostly below 1, whereas elevated scores were observed in patients with LGC. Compared with Asian patients with LGC, Caucasian patients displayed a more pronounced increase in levels, which was primarily attributed to the higher proportion of advanced-stage cases among Caucasians. We only analyzed healthy African-American and Hispanic people, and their scores were also below 1, similar to those of Asians and Caucasians. These findings suggest that the results of the CSGSA method for lung cancer are consistent across different ethnicities.

### 3.9. Method Validation

To confirm the reproducibility of glycopeptide analysis, the process from pretreatment to LC-MS measurement was repeated five times daily, using serum from a cancer patient, over three consecutive days. Both intra- and inter-day reproducibility were assessed. For intra-day measurements, the coefficient of variation (CV) was calculated from five measurements taken on the first day. For inter-day reproducibility, daily averages were computed, and the CV across three days was determined. The intra-day CVs of AT271-FSG and MG70-FSG were 4.8% and 9.1%, respectively, while the inter-day CVs were 3.3% and 7.5%, respectively. More than 80% of the EGPs demonstrated a CV of less than 30% (Figure 6B). In previous studies, we evaluated the stability of the collection tube, stability after serum separation, the influence of diet (diurnal variation), and inter-institutional and inter-instrumental differences, confirming that none of these factors affected the results [27].

### 3.10. Key Contributors to Model Efficacy

Identifying the factors that play a crucial role in constructing a neural network model is the key to gaining a better understanding of lung cancer detection; however, the neural network framework significantly limits the visibility with which explanatory variables influence the model’s outcomes. In contrast, XGBoost allows for the identification of such contributions; therefore, we utilized it to estimate the contributing factors, although they were not direct contributors. In XGBoost, the ‘F-score’ indicates the frequency with which a feature is used to split the data across all trees in the model. A higher F-score suggests that the feature plays a more significant role in creating a model, which is crucial for the model’s decision-making process. Our results showed that CEA and CYFRA made high contributions, which was corroborated by their strong ROC-AUC performance (Figure 1A). SCC antigen also showed a high score, suggesting that it plays a significant role in distinguishing squamous cell carcinoma. Among the PCA-derived features, PC2 and PC3 were particularly influential (Figure 6C).

## 4. Discussion

In this study, we developed a novel screening method that integrates cancer-specific tumor markers and glycan alterations for the early detection of lung cancer. Our approach revealed that α1-antitrypsin with fully sialylated biantennary glycan attached to Asn 271 (AT271-FSG) and α1-macroglobulin with fully sialylated biantennary glycan attached to Asn 70 (MG70-FSG) can significantly distinguish between patients with LGC and healthy controls. Numerous studies have documented that α2-macroglobulin and α1-antitrypsin undergo glycosylation changes upon the onset of cancer. For example, Šunderić et al. observed the significant elevation of α2-macroglobulin with N-glycans, including α2,6 sialylation, N-acetylglucosamine residues, and tri-/tetraantennary high-mannose-type complexes, in patients with colorectal cancer [38]. Additionally, Mondal et al. reported that sialylation, fucosylation, and high-glycan branching on α1-antitrypsin glycans are significantly elevated in patients with hepatocellular carcinoma [39]. Our study is the first to demonstrate that these glycosylation changes occur at specific sites on these proteins during the onset of lung cancer. Additionally, we achieved ROC-AUC scores of up to 0.935 by employing machine learning to synthesize data from nine tumor markers, two glycopeptides, and 1688 EGPs, demonstrating superior diagnostic accuracy. It also achieved an ROC-AUC of 0.914 when comparing the stage I lung cancer group with healthy individuals, suggesting that the CSGSA method may effectively detect early-stage cancer. When setting the cutoff values, priority was given to increasing both the specificity and sensitivity. This approach was adopted because while correctly identifying cancer patients is crucial, it is equally important in practice to minimize false positives, particularly in screening programs where a small number of cases are detected within a large healthy population. For example, when specificity drops below 95%, the PPV for many cancer types falls below 1%. Even with high sensitivity, a PPV below 1% means that over 99% of positive results are false positives, leading to unnecessary anxiety for both the examinee and the physician.

In this study, we achieved a prevalence-corrected PPV of 2.8% and a negative predictive value (NPV) exceeding 99.9%; the PPV was significantly higher than the industry benchmark of <1% [40]. Compared with other newly developed biomarkers, such as microRNAs [10] or tumor-secreted proteins [9], our method, CSGSA, offers distinct advantages in terms of accuracy, cost-effectiveness, and reproducibility, indicating the reliability of this method as a screening tool. This method can potentially reduce the use of more invasive procedures, such as computed tomography.

However, this study had some limitations. (i) As this was a retrospective study involving patients already diagnosed with lung cancer, it remains uncertain whether CSGSA can effectively identify asymptomatic patients with LGC. To validate these findings, a longitudinal, prospective approach is required. In recent years, numerous cohort studies have investigated long-term serum banking and regular physical examinations of residents in specific areas. Longitudinal studies using these samples would be valuable for assessing the effectiveness of early detection. (ii) It is crucial to evaluate the ability of this method to distinguish lung cancers from other types of cancer, such as colorectal, gastric, liver, prostate, and pancreatic cancers. Additionally, it is important to compare lung cancer with other cancers and also benign lung diseases. In this study, we observed a slight increase in CSGSA scores among six patients with benign lung diseases, highlighting the importance of statistically evaluating the differences between lung cancer and benign lung disease groups. (iii) A sufficient number of cases is required to optimize the machine learning model; however, the dataset in the present study was not large enough. Future research should focus on expanding the dataset and exploring additional glycan markers to enhance the accuracy of lung cancer diagnosis. (iv) This study did not identify all glycopeptides that contributed to the differentiation between healthy individuals and cancer patients. Unlike proteomics, which focuses solely on identifying protein species, glycoproteomics requires both the identification of proteins and the determination of glycosylation sites and glycan structures. Additionally, although proteomics benefits from comprehensive MS/MS libraries and databases, no equivalent resources exist for glycopeptides, owing to the complexity of their MS/MS patterns. Although we successfully identified several glycopeptide structures as cancer markers in previous studies [25,26], these efforts were time-consuming and labor-intensive. Given these challenges, we chose not to exhaustively identify all 1688 glycopeptides in this study, instead prioritizing the processing of a large number of samples. However, we anticipate that ongoing updates to glycopeptide databases and improvements in mass spectrometry sensitivity will facilitate the identification of additional glycopeptides. (v) The PPV of 2.8% needs to be further improved for CSGSA to become a reliable and practical screening method. In this study, separate analyses of adenocarcinoma and squamous cell carcinoma were not conducted because the number of cases was insufficient for machine learning. However, because these two cancer types exhibit different characteristics, developing separate machine-learning models for each histological type could be a viable approach to achieving a higher PPV.

## 5. Conclusions

The neural network model used in this study, CSGSA, successfully discriminated patients with LGC from healthy controls, with an impressive ROC–AUC of 0.935, outperforming existing tumor markers. It also identified stage I cases with an ROC-AUC of 0.914, indicating the possibility of early-stage detection. The PPV reached 2.8%, which was sufficient for practical application. This approach promises to revolutionize lung cancer diagnosis by identifying the onset of lung cancer in asymptomatic individuals.

## 6. Patents

LSI Medience Corporation applied for a patent related to this research in Japan (Japanese Patent Application No. 2023-105968).

## Figures and Tables

**Figure 1 cancers-17-01474-f001:**
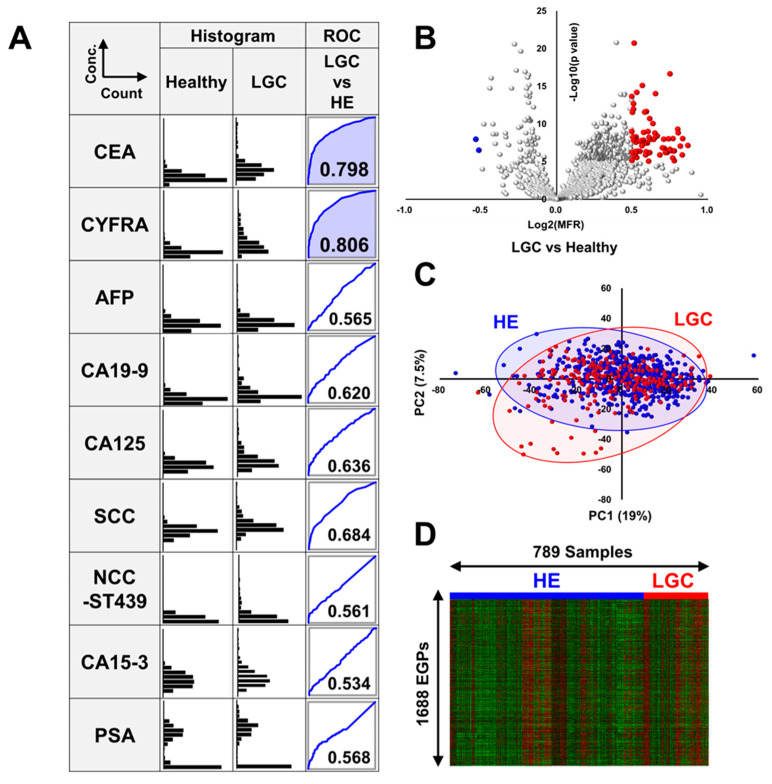
Levels of tumor markers, volcano plot analysis, principal component analysis, and heatmap analysis. (**A**) This figure depicts the levels of nine tumor markers—CEA, CYFRA, AFP, CA19-9, CA125, SCC antigen, NCC-ST439, CA15-3, and PSA—in patients with LGC and healthy volunteers. Histograms illustrate the distribution of logarithmically transformed marker levels (log base 10) on the vertical axis against the number of individuals on the horizontal axis. Additionally, ROC curves compare the diagnostic performance of each marker between the LGC and healthy groups, with the area under the curve (AUC) values indicated. Curves demonstrating AUC values over 0.7 are highlighted with blue shading. (**B**) A volcano plot comparing patients with LGC with healthy individuals. The vertical axis represents the negative logarithm base 10 of the *p*-value derived from Student’s *t*-test, and the horizontal axis represents the logarithm base 2 of the mean-fold ratio (MFR). EGPs with a *p*-value less than 10^−10^ and an MFR greater than 2^0.5^ are highlighted in red, whereas those with a *p*-value less than 10^−10^ and an MFR less than 2^−0.5^ are highlighted in blue. All other points are represented by gray dots. (**C**) A score plot from principal component analysis (PCA) showing data for LGC (red) and healthy (blue) groups. (**D**) Heatmap displaying expression profiles of 1688 EGPs across 789 individuals. Overexpressed EGPs are shown in red and downregulated EGPs are shown in green.

**Figure 2 cancers-17-01474-f002:**
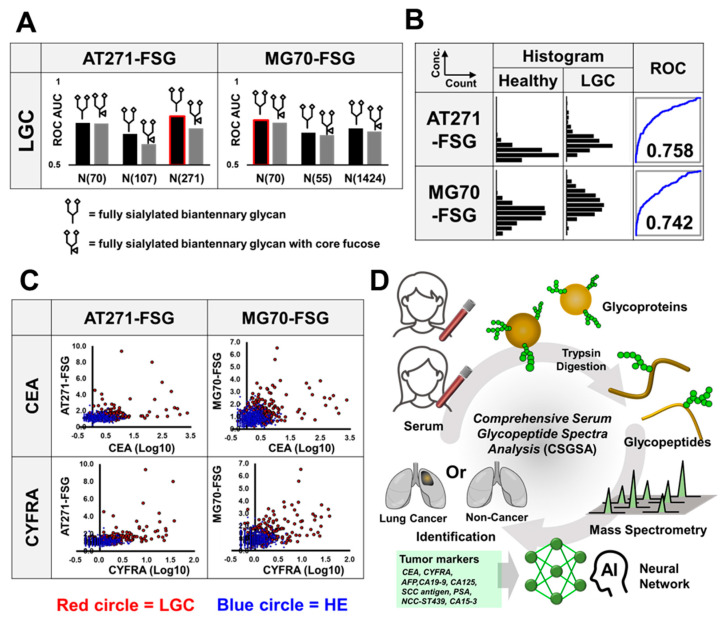
Analysis of α1-antitrypsin and α2-macroglobulin-derived glycopeptides. (**A**) ROC analysis was conducted between the healthy and LGC groups, and the AUC values for glycopeptides derived from antitrypsin (AT271-FSG) and macroglobulin (MG70-FSG) are illustrated. Black bar: ROC-AUC when the FSG is a fully sialylated biantennary glycan. Gray bar: ROC-AUC when the FSG is a fully sialylated biantennary glycan with core fucose. The glycopeptides used in the machine learning model are highlighted with red borders. Numbers following “N” indicate the position of the asparagine residue from the N-terminus. (**B**) Histograms displaying the distribution of AT and MG glycopeptide levels across lung cancer (LGC) and healthy (HE) groups. Accompanying ROC curves illustrate the diagnostic accuracy of each glycopeptide, with AUC values indicated for differentiation between cancer and healthy groups. (**C**) Scatter plots showing the correlation between glycopeptide levels (AT271-FSG and MG70-FSG) and tumor markers (CEA and CYFRA). The vertical axes represent the levels of AT271-FSG or MG70-FSG, while the horizontal axes show the logarithmically transformed levels of the tumor markers. (**D**) Comprehensive Serum Glycopeptide Spectra Analysis (CSGSA): An illustration of a lung cancer detection model combining 1688 EGPs and nine tumor markers.

**Figure 3 cancers-17-01474-f003:**
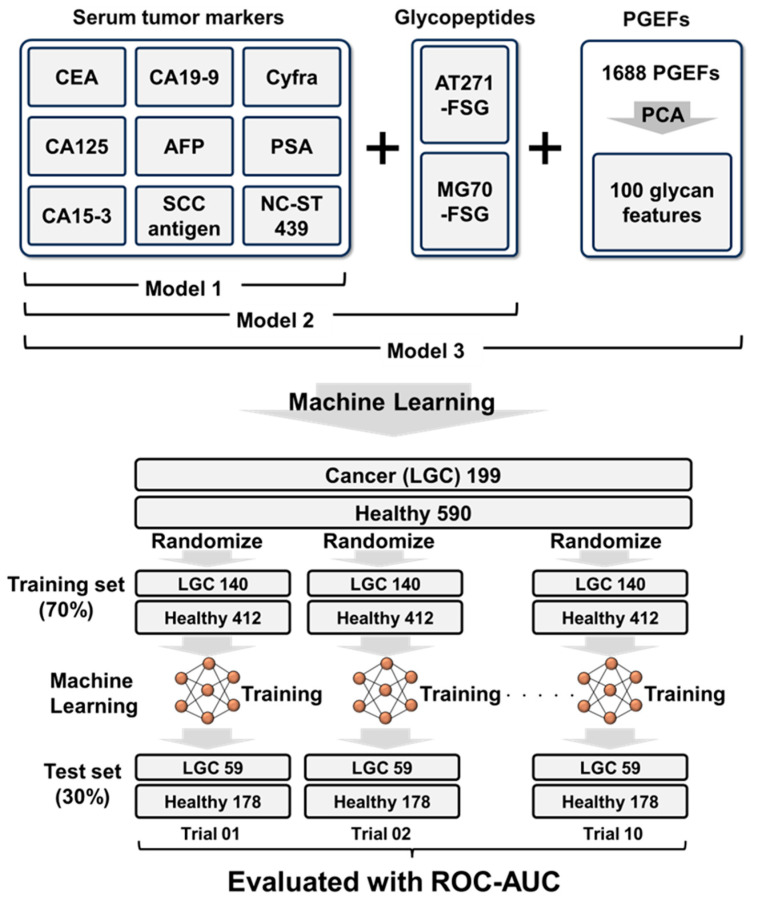
The machine learning model development and evaluation strategy. This figure outlines the development and evaluation of three distinct machine learning models to assess the impact of different biomarker sets on cancer diagnosis accuracy. Model 1 incorporates nine conventional serum tumor markers, Model 2 includes two glycopeptides, AT271-FSG and MG70-FSG, and Model 3 integrates these components with 100 key glycan features derived from a principal component analysis (PCA) of 1688 EGPs. Each model was trained using a randomly selected training set comprising 70% of the samples, with the remaining 30% serving as the test set. The evaluation process was repeated 10 times to validate consistency, and results were collectively analyzed through ROC analysis to measure performance across various configurations.

**Figure 4 cancers-17-01474-f004:**
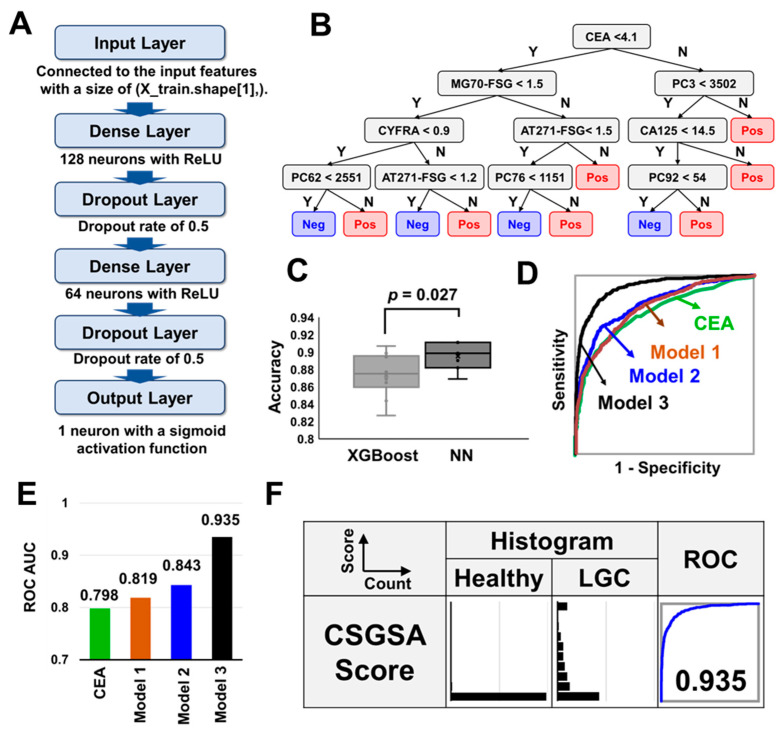
A comprehensive evaluation of machine learning models and CSGSA score. (**A**) The structure of the neural network model: The neural network consists of an input layer, followed by two sets of dense and dropout layers, and finally an output layer. Dropout layers play a role in preventing overfitting by randomly setting a fraction of input units to zero during training. (**B**) An example of a tree structure optimized by XGBoost: the actual tree structure, particularly in the lower nodes, changes each time depending on the selection of the training set. (**C**) Model comparison: performance comparison of cancer identification using Model 3 with XGBoost and Neural Network (NN) algorithms, showcasing accuracy. (**D**) ROC curves: displays the ROC curves for three different models. (**E**) Model performance: ROC-AUC values for each model, illustrating their effectiveness in cancer identification. (**F**) CSGSA score distribution and performance: Histograms showing the distribution of CSGSA scores across LGC and healthy groups. Adjacent ROC curves evaluate the diagnostic performance of Model 3.

**Figure 5 cancers-17-01474-f005:**
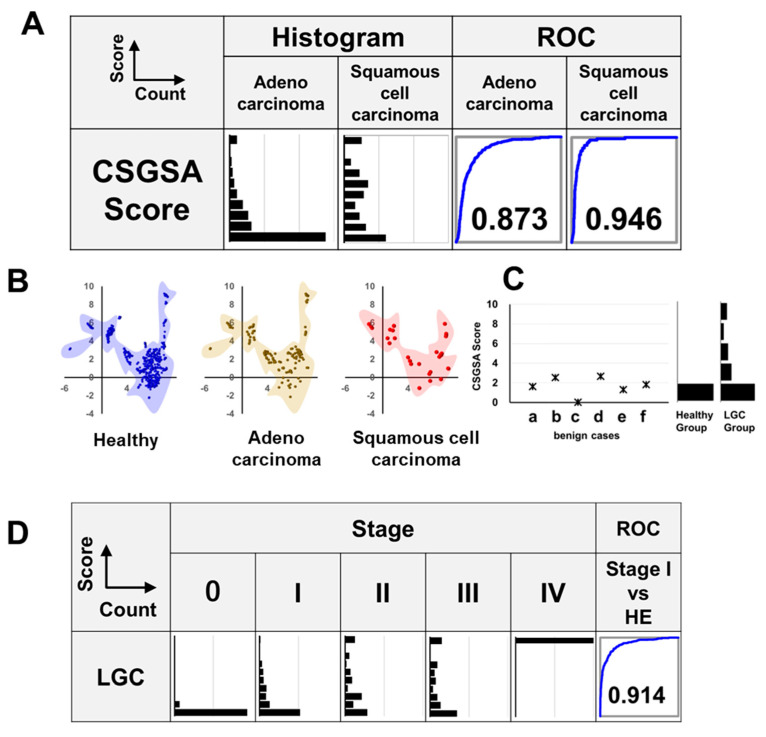
Evaluation of CSGSA: impact of lung cancer histology, comparison to benign lung disease, and relationship to stage of cancer progression. (**A**) Relationship between lung cancer histological types (adenocarcinoma and squamous cell carcinoma) and CSGSA scores: histogram and ROC curves (compared with the healthy control group). (**B**) UMAP analysis of patients with adenocarcinoma, squamous cell carcinoma, and healthy individuals. (**C**) CSGSA determination for patients with benign lung disease: (a) benign tumor, (b) inflammatory lung disease, (c) pulmonary fibrosis, (d) pulmonary fibrosis, (e) benign tumor, and (f) pulmonary fibrosis. Each case is shown alongside reference histograms of scores from healthy individuals and lung cancer (LGC) patients for comparison. (**D**) Staging analysis: histograms displaying the distribution of CSGSA scores across cancer stages. ROC analysis compares stage I against the healthy group, indicating early detection capability.

**Figure 6 cancers-17-01474-f006:**
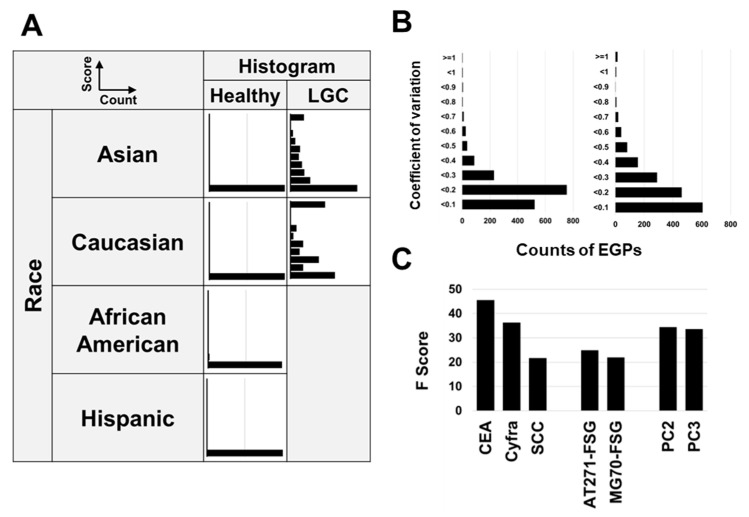
Evaluation of CSGSA: effects of racial differences, measurement reproducibility, and influence factors on the machine learning model. (**A**) Relationship between CSGSA scores and ethnicity. Histograms depict CSGSA scores across healthy individuals, and patients with LGC in Asian and Caucasian populations. African-American and Hispanic groups are only represented in histograms with respect to healthy individuals. (**B**) Intra-day and inter-day reproducibility of 1688 EGPs: Each EGP was measured five times within the same day (including preparation and MS measurement errors), and the coefficient of variance (CV) for each EGP was calculated and presented as a histogram. For the inter-day reproducibility, measurements were taken five times per day, and the averages of these values were obtained. This process was repeated over three days, and the variation (CV values) of the mean values (*n* = 3) are shown in a histogram. (**C**) Contributors to Model 3 were identified through XGBoost’s F-scores, which visualized the contribution of each factor to the model.

**Table 1 cancers-17-01474-t001:** Demographic characteristics of the patients.

Condition	Age	Number	Sex (Man Ratio)	Stage	Race
Healthy Volunteers (HE)	48.2 (±12.2)	590	50.3%		Asian (297) Caucasian (114) African American (115) Hispanic (63) Mixed Ethnicities (1)
Lung Cancer (LGC)	68.8 (±9.7)	199	57.4%	Stage 0 (5) Stage I (124) Stage II (19) Stage III (20) Stage IV (2) Unclassified (29)	Asian (179) Caucasian (20)
Total	53.3 (±12.2)	789	52.1%		

**Table 2 cancers-17-01474-t002:** Evaluation of the cancer screening model. Patients with lung cancer and healthy individuals.

			True State			
			LGC	Healthy	Sum	PPV or NPV
Predicted State	Observed Samples	Positive	336	35	371	
		Negative	251	1748	1999	
		Sum	587	1783	2370	
	Prevalence Correction	Positive	57	1961	2018	2.8%
		Negative	43	97,939	97,982	99.96%
		Sum	100	99,900	100,000	
	Sensitivity or specificity		57.2%	98.0%		

## Data Availability

The levels of nine tumor markers, AT271-FSG and MG70-FSG glycopeptides, and EGPs expression, along with anonymized case information, are ready for disclosure. The data will be accessible after publication. Researchers seeking raw data should contact the corresponding authors via email. Access requires approval from each institution’s ethics committee and a collaborative agreement with the corresponding authors.

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
