# Peer review of "Comprehensive Serum Glycopeptide Spectra Analysis Combined with Machine Learning for Early Detection of Lung Cancer: A Case–Control Study"

_cancers, 2025, doi:10.3390/cancers17091474_

Round 1

Reviewer 1 Report

Comments and Suggestions for Authors

Comments and Suggestions for Authors:

  1. The ML learning model XGBoost performed well when combined with nine biomarkers and two glycopeptides. However, the results without the combination were not presented separately.
  2. Figure 4 illustrates A (MLP) and B (XGBoost). I'm not clear how they integrate them. A separate figure for D is expected.
  3. The author(s) only used XGBoost. Why not Random forests or Extra Decision Trees? What makes a difference? How many input features from each patient feed into the XGBoost train and test?
  4. The ROC-AUC of ML and CSGSA can be demonstrated separately to see the effectiveness.
  5. ML and CSGSA scores comparison using only six patients does not make strong support for this study.
  6. Line 78, which provides a review reference, and I'm referring to the latest application, https://doi.org/10.1117/12.3045828

Reviewer 2 Report

Comments and Suggestions for Authors

The manuscript presented by Yamazaki et al.  proposes an innovative approach: integrating the search for cancer-specific tumor markers with glycan alterations as a new method for the early detection of lung cancer. This method identifies α1-antitrypsin with a fully sialylated biantennary glycan at Asn 271 (AT271-FSG) and α1-macroglobulin with the same glycan at Asn 70 (MG70-FSG) as factors with significant differences between lung cancer patients and healthy individuals. 
Minor comments:
The study design limits its ability to confirm efficacy in detecting asymptomatic lung cancer patients. Please indicate the limitation in the discussion section and suggest a possible strategy to mitigate it (i.e., conduct a prospective longitudinal study, which is required for validation).
It seemed that the machine learning model could not be adequately optimized using the dataset that was provided. The robustness of the model would be enhanced by increasing the sample size. Please also discuss this point.

Reviewer 3 Report

Comments and Suggestions for Authors

Title: Comprehensive serum glycopeptide spectra analysis combined with machine learning for early detection of lung cancer: a case-control study

Summary: In this study, authors developed machine learning models to improve diagnosis of lung cancer to achieve better sensitivity than single biomarkers. Authors showed their model could diagnose squamous cell carcinoma better than Adenocarcinoma. Their model also displayed high sensitivity to diagnose early-stage LC.

Overall, the study is very well designed and can be very useful in more sensitive lung cancer diagnosis.

Major comments:

  1. Authors should provide the rationale behind choosing the 9 tumor markers. What was the basis of this selection.
  2. Fig 2: Since the first heatmap did not show any differences between the 2 groups, Authors should show the heatmap with the threshold values applied for p value and fold change that led to identification of differences between healthy and lung cancer group.
  3. Fig 2A: Figure is not clear. Please label and explain, what do the black and grey bars represent? What are the comparison groups? If you are comparing black and grey bars, please provide p values from statistical analysis.
  4. Fig 2A: Please show comparison between healthy and LC samples.
  5. Fig 5C: Authors should plot CSGSA score for healthy, Adeno and squamous samples alongside the 6 lung diseases samples.

Author Response

Dear Reviewer,

We would like to express our sincere gratitude for the time and effort you devoted to evaluating our manuscript entitled "Comprehensive serum glycopeptide spectra analysis combined with machine learning for early detection of lung cancer: a case-control study."

Your thoughtful and constructive feedback has been invaluable in improving the quality and clarity of our work. We have carefully reviewed all of your comments and have addressed each point below. All corresponding revisions in the manuscript are highlighted in yellow.

Comment 1:

Authors should provide the rationale behind choosing the 9 tumor markers. What was the basis of this selection.

Response 1:

Thank you very much for your valuable comment regarding the selection of tumor markers. We have revised the manuscript to clarify the rationale (line 189). Although the primary objective of this study is the early detection of lung cancer, we deliberately selected a panel of nine tumor markers with the intention of enabling future application of this diagnostic framework to other major malignancies, including ovarian, gastrointestinal, pancreatic, prostate, breast, and liver cancers. These markers were chosen based on the following considerations: CEA: colorectal, gastric, lung, pancreatic, CYFRA: non-small cell lung cancer (especially squamous cell type), CA19-9: pancreatic and gastrointestinal cancers, CA125: ovarian and other gynecological cancers, AFP: hepatocellular carcinoma, NCC-ST-439: gastrointestinal and lung cancers, PSA: prostate cancer, SCC-antigen: squamous cell carcinomas and CA15-3: breast cancer. All of these tumor markers are covered by the Japanese national health insurance system and are routinely measured in standard clinical practice. Their inclusion enhances the practical applicability and scalability of our model in real-world healthcare settings without the need for additional infrastructure.

Comment 2:

Fig 2: Since the first heatmap did not show any differences between the 2 groups, Authors should show the heatmap with the threshold values applied for p value and fold change that led to identification of differences between healthy and lung cancer group.

Response 2:

Thank you very much for your comment regarding Figure 1D (You indicate Heatmap as Figure 2, which I will answer below as Figure 1D).

We understand your suggestion to apply threshold values for p-values and fold change to highlight the glycopeptides that differ significantly between healthy individuals and lung cancer patients. However, doing so would result in a figure nearly identical to the volcano plot already shown in Figure 1B. Instead, the purpose of Figure 1D (as well as the PCA plot in Figure 1C) was to demonstrate that the majority of enriched glycopeptides (EGPs) do not show substantial differences between the two groups. This suggests that only a small subset of EGPs undergo changes associated with the onset of lung cancer.

By presenting these three complementary visualizations—(i) the volcano plot in Figure 1B, (ii) the PCA in Figure 1C, and (iii) the heatmap in Figure 1D—we aimed to provide a holistic view: although some EGPs do change significantly (Figure 1B), most remain stable (Figures 1C and 1D). This highlights the importance of focusing on a select group of discriminative features when constructing a diagnostic model. We hope this rationale clarifies our intention in presenting the original heatmap without statistical filtering.

Comment 3:

Fig 2A: Figure is not clear. Please label and explain, what do the black and grey bars

represent? What are the comparison groups? If you are comparing black and grey

bars, please provide p values from statistical analysis.

Response 3:

Thank you very much for your thoughtful comment. In Figure 2A, we present ROC analyses comparing glycopeptides derived from antitrypsin and macroglobulin between the lung cancer and healthy groups. The bar graph shows the AUC values resulting from these analyses. Specifically, the black bars represent glycopeptides with a fully sialylated biantennary glycan, while the grey bars represent glycopeptides with a fully sialylated biantennary glycan containing core fucose. To improve clarity, we have revised the figure legend accordingly (line 345). We are not statistically comparing the black and grey bars directly, but rather using them to illustrate the classification performance (AUC) of glycopeptides with different glycan structures.

We appreciate your suggestion and hope the revised legend provides a clearer explanation.

Comment 4:

Fig 2A: Please show comparison between healthy and LC samples.

Response 4:

Same as above reply.

Comment 5:

Fig 5C: Authors should plot CSGSA score for healthy, Adeno and squamous samples

alongside the 6 lung diseases samples.

Response 5:

Thank you for your insightful comment. As you suggested, we agree that showing the CSGSA scores for healthy individuals and lung cancer patients alongside those of the benign lung disease cases improves clarity. Since we have a large number of healthy and lung cancer samples, we chose to present their scores as histograms next to the individual benign cases. The figure legend has also been revised accordingly (line 455).

Sincerely,

Kazuhiro Tanabe (on behalf of all co-authors)

Round 2

Reviewer 3 Report

Comments and Suggestions for Authors

Authors have addressed all concerns. Manuscript looks good for publication.